# OptAgent: Optimizing Query Rewriting for E-commerce via Multi-Agent Simulation

## Abstract

Deploying capable and user-aligned LLM-based systems necessitates reliable evaluation. While LLMs excel in verifiable tasks like coding and mathematics, where gold-standard solutions are available, adoption remains challenging for subjective tasks that lack a single correct answer. E-commerce Query Rewriting (QR) is one such problem where determining whether a rewritten query properly captures the user intent is extremely difficult to figure out algorithmically. In this work, we introduce OptAgent, a novel framework that combines multi-agent simulations with genetic algorithms to verify and optimize queries for QR. Instead of relying on a static reward model or a single LLM judge, our approach uses multiple LLM-based agents, each acting as a simulated shopping customer, as a dynamic reward signal. The average of these agent-derived scores serves as an effective fitness function for an evolutionary algorithm that iteratively refines the user's initial query. We evaluate OptAgent on a dataset of 1000 real-world e-commerce queries in five different categories, and we observe an average improvement of 21.98% over the original user query and 3.36% over a Best-of-N LLM rewriting baseline.

## 1 Introduction

Large language models (LLMs) are increasingly being used as agents to automate complex tasks (ZHAO et al., 2023; Chen et al., 2024c), with capabilities enhanced by additional test-time computation (Zhang et al., 2025; Muennighoff et al., 2025). Their success has been most pronounced in verifiable domains like mathematics (Shao et al., 2024), coding (Tang et al., 2024), scientific discovery (Kumbhar et al., 2025), reasoning (RRV et al., 2025; Wang et al., 2024), and planning (Parmar et al., 2025), where the correctness of an output can be unambiguously determined. This verifiability provides a crisp reward signal, enabling powerful optimization techniques like Self-Taught Reasoner (STaR) (Zelikman et al., 2022) and Group Relative Policy Optimization (GRPO) (Shao et al., 2024).

However, this paradigm breaks down in numerous real-world applications, such as e-commerce Query Rewriting (QR), where the goal is to reformulate user queries to match their latent intent, e.g., faster shipping or high-quality reviews. Platforms like Amazon and Etsy process millions of user queries on a daily basis[1], where the user's query is often short, ambiguous, or riddled with typos. In such domains, the absence of a gold-standard solution renders existing optimization methods ineffective. While Reinforcement Learning from Human Feedback (RLHF) (Ouyang et al., 2022) utilizes expert annotation, its prohibitive cost and slow pace make it impractical for rapid, iterative optimization. Techniques such as Reinforcement Learning from AI Feedback (RLAIF) (Lee et al., 2023) replace this labor-intensive expert annotation with an LLM to evaluate the responses. While promising, using the "LLM-as-a-Judge" approach is not without its limitations (Li et al., 2023b). A growing body of research has revealed that a single LLM judge is prone to significant biases (e.g., position, verbosity) (Gallegos et al., 2024), a lack of robustness (Chen et al., 2024b), and can be unreliable (Tian et al., 2023), especially when evaluating complex, multi-faceted criteria.

In this paper, we show that the quality of a solution in such subjective domains is better approximated not by a single judge, but by a dynamic, simulation-based evaluation. We present an ensemble of

---

[1] https://redstagfulfillment.com/how-many-daily-visits-does-amazon-receive
https://www.yaguara.co/etsy-statistics/

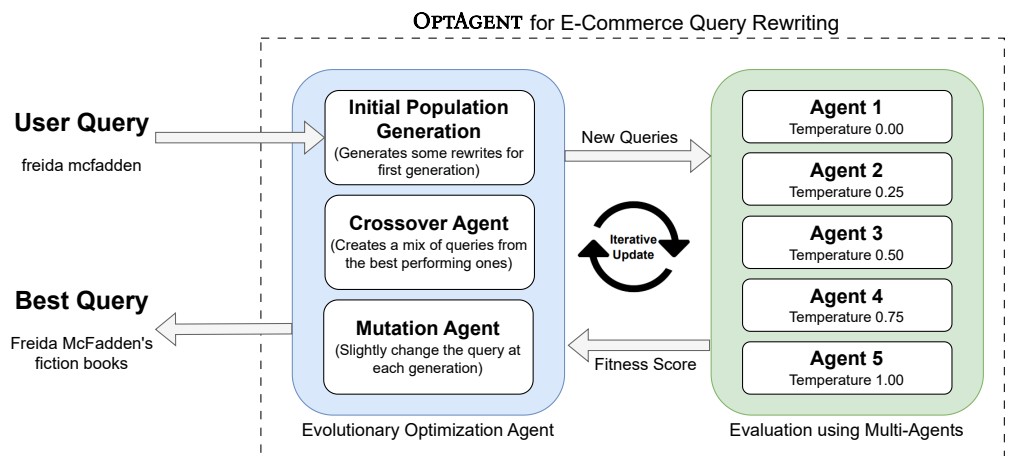

Figure 1: An overview of the OPTAGENT framework in query rewriting for e-commerce applications. The user's initial query is passed to the framework, where we first populate the initial generation with candidate rewrites and then perform evolutionary optimization to search for better query rewrites. The fitness of each candidate is determined by its performance in a multi-agent simulation populated by LLM-based shopper agents.

LLM-based agents, acting as simulated users, that can produce a more robust and nuanced reward signal. We introduce diversity into this agent population through the simple and effective mechanism of temperature sampling (Balachandran et al., 2025; Renze, 2024). By instantiating each agent with a different temperature, we encourage a variety of reasoning paths and evaluation perspectives, allowing the ensemble's collective judgment to form a rich and multi-faceted evaluation that better approximates the complexity of true human preference.

Existing QR methods often require vast amounts of historical user interaction data, which may not be available for new or infrequent ("tail") queries, and they struggle to optimize for the latent, subjective quality of such a rewrite. Moreover, recent works (Song et al., 2025; Li et al., 2025) have shown that RL-based techniques, e.g., GRPO, fail to properly generate diverse outputs, which limits their performance. On the other hand, evolutionary algorithms have been shown to outperform these RL-based methods with less compute (Agrawal et al., 2025).

Motivated by these insights, we present OPTAGENT, an agentic framework that couples our agent simulation-based rewards with evolutionary algorithms to optimize queries in QR. OPTAGENT begins by generating a population of candidate solutions and iteratively refining them by leveraging LLMs to perform the genetic operators of crossover and mutation directly in natural language, enabling a semantically aware exploration of the solution space. The fitness of each query is determined by its performance in the multi-agent simulation. Our experiments on a dataset of 1000 real user queries show that OPTAGENT successfully improves the relevance of products by 21.98% over the user's original query and outperforms the established baseline of Best-of-N (BoN) using LLMs by 3.36%, with the largest improvement coming from *tail queries* (4.50% improvement over BoN).

***Contributions.*** Our contributions are as follows:

1. We design an evaluation mechanism where a multi-agent simulation, populated by LLM agents with diverse reasoning styles (via temperature sampling), serves as the fitness function for an evolutionary algorithm.

2. We introduce OPTAGENT, a novel framework that optimizes outputs in e-commerce query rewriting by replacing static reward functions with a dynamic score derived from a multi-agent simulation.

3. We provide an empirical validation of OPTAGENT demonstrating a 21.98% improvement in relevance over user queries and outperforming the Best-of-N LLM baseline by 3.36%.

## 2 RELATED WORKS

**LLMs as a Judge.** With their ability to process natural language and use their internal world models (Gu et al., 2024; Ge et al., 2024), LLMs have presented a compelling alternative to the traditional expert-driven evaluation (Li et al., 2023a; Zhu et al., 2023). They have achieved remarkable success evaluating responses across diverse domains, ranging from text-generation (Badshah & Sajjad, 2024; Zheng et al., 2023), finance (Brief et al., 2024; Yu et al., 2024), or law (Ma et al., 2024; Cheong et al., 2024). Moreover, LLMs have become sufficiently flexible to handle multi-modal inputs (Khattak et al., 2023) and can evaluate multi-modal responses as well (Chen et al., 2024a; Wu et al., 2024). However, several works have shown these LLM-based evaluators to have bias (Gallegos et al., 2024; Tan et al., 2024), lack of reliability (Tian et al., 2023), and lack of robustness (Chen et al., 2024b; Handa et al., 2024). While human judges also exhibit inherent bias (Wu & Aji, 2023; Parmar et al., 2022; Clark et al., 2021), they are more reliable, especially when multiple judges evaluate the same problem. Inspired by this, multiple works have used an ensemble of LLMs, or test-time algorithms to judge a single query, instead of relying on a single LLM (Bermejo, 2024; Jiang et al., 2024). We use this insight to build multiple multi-modal LLM-based agents that analyze each query independently, and we then aggregate their scores to get the final score.

Additionally, several works (Brown et al., 2024; Wang et al., 2025b) have shown that using the same LLM multiple times can output similar responses. Therefore, we vary the temperature for each agent, a technique known as temperature sampling (Renze, 2024), to ensure different reasoning paths emerge for each agent. Most importantly, we verify the insight from Liu et al. (2023b) that fine-grained continuous scores can be achieved from the weighted average of the discrete scores. In this work, we demonstrate this by using each agent to assign a semantic score, i.e., a classification of a product as "Fully Relevant", "Partially Relevant", or "Irrelevant" for a given query.

**Evolutionary Methods.** OPTAGENT extends a long tradition of research on evolutionary or genetic programming (Langdon & Poli, 2013), where one repeatedly uses a set of mutation and crossover operators to evolve a pool of queries or prompts (Yang et al., 2023; Liu et al., 2023a; Agrawal et al., 2025). Recently, these classical algorithms have been supercharged by LLMs, which can operate directly on natural language. For instance, evolutionary algorithms have succeeded in symbolic regression applications (Ma et al., 2022; Schmidt & Lipson, 2009), automated discovery (Novikov et al., 2025; Cranmer, 2023; Chen et al., 2023), and scheduling (Zhang et al., 2021) problems. However, a challenge with these methods is the reliance on automated and often objective evaluation methods, which can be extremely difficult to design for several real-world domains. In contrast, OPTAGENT leverages a multi-LLM-based agentic simulation for its fitness function.

**Query Rewriting for E-Commerce.** Query rewriting (QR) has emerged as a critical, indispensable component of modern e-commerce search engines. Its primary function is to refine, reformulate, and enhance ambiguous or incomplete customer queries into well-formed inputs that can be more effectively processed by a search and retrieval system. Historically, QR has been approached by matching with historical data with the use of encoder-only models (Li et al., 2022) instead of generating new queries. Meanwhile, work done by Zhang et al. (2022) has used Seq2Seq models for semantic classification. With the rise and potential of LLMs, several works have started to integrate LLMs into their systems. Agrawal et al. (2023) and Dai et al. (2024) have used reinforcement learning to generate query reformulations by training a generative model to directly optimize the semantic similarity. Further improvements include using a "session graph" of a user's search history to provide a more customizable match based on historic data (Zuo et al., 2022). However, these methods often require vast amounts of historical user interaction data, which may not be available for new or infrequent ("tail") queries, and they struggle to optimize for the latent, subjective quality of such a rewrite. Our work addresses this gap by proposing a method that does not rely on historical logs for optimization but instead simulates user preferences to guide the search for better queries, making it particularly well-suited for the long tail of search queries.

## 3 OPTAGENT

OPTAGENT is a framework for optimizing Query Rewriting (QR), a domain where evaluation is extremely challenging due to the subjective and latent nature of human preferences. In this section,

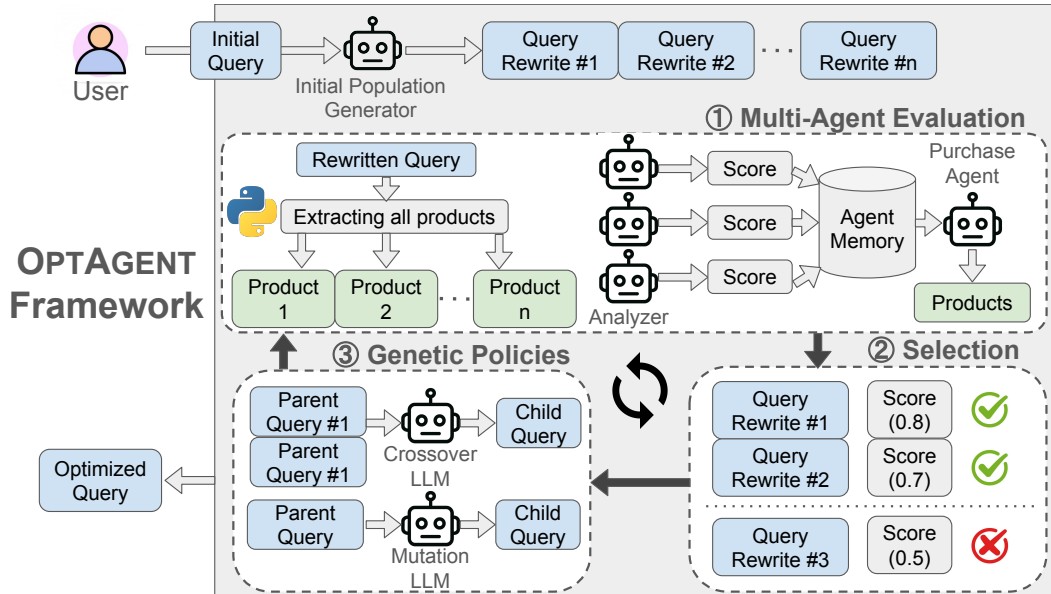

Figure 2: Overview of our OPTAGENT framework for e-commerce query rewriting application. The blue blocks represent the queries, and the green blocks represent the products listed on the platform. The user's initial query is first rephrased multiple times by an LLM, which acts as the initial population for our evolutionary framework. The following steps are repeated until the computation budget is exhausted: 1) Each query is evaluated by our multi-agent simulation, where each agent analyzes all products and stores their semantic scores in the memory. Then the purchase agent loads all the products and decides which ones to purchase, along with the total cost. 2) The semantic scores and the total amount spent constitute the final fitness function for the given query. Top $N$ queries are passed to the next generation and used as parents to populate the next generation. 3) New queries are generated via crossover (mixing two parent queries) and mutation (altering a single parent query).

we detail the two primary components of our framework: the multi-agent simulation, which serves as a fitness function (§3.1) and the genetic algorithm that drives the optimization (§3.2). Figure 2 illustrates the OPTAGENT framework for QR in more detail.

## 3.1 EVALUATION USING MULTI-AGENT SIMULATION

Traditionally, the quality of the rewritten query in QR is evaluated by measuring the semantic relevance of the rewritten and the original query (Rokon et al., 2024; Liu et al., 2022). Semantic Relevance involves classifying a product into "Fully Relevant", "Partially Relevant", or "Irrelevant" for the given query. Inspired by the rise of the LLM-as-a-judge paradigm, recent works (Sachdev et al., 2024; Chaudhary et al., 2023) have used LLMs for classifying semantic relevance. Moreover, recent works have integrated a persona to LLM agents, simulating real-world users (Lu et al., 2025; Wang et al., 2025a). While these agents exhibit diverse reasoning paths based on their personas, we found that they also fall into inherent biases, which are undesirable. We detail a case study in Appendix C.1, which shows how even a benign personality trait can lead to discriminatory behaviors.

**Temperature Sampling** To overcome persona biases and generate diverse reasoning paths, we take inspiration from advances in test-time algorithms and implement an ensemble of agents, $\mathcal{A} = \{a_1, a_2, ..., a_K\}$, where $K$ is the number of agents. Each agent, $a_i \in \mathcal{A}$, is initialized not with a persona but with a unique sampling temperature $T_i$. A low temperature makes the distribution sharper, favoring the most probable tokens and leading to more deterministic outputs. A higher temperature flattens the distribution, increasing the probability of sampling less likely tokens. This encourages the model to explore different, but still plausible, paths of reasoning and expression without injecting a pre-defined persona. This is known as temperature sampling (Renze, 2024).

While this approach doesn't address the implicit, foundational biases of the base model, it does reduce the bias generated due to the persona.

**Agent Simulation** For a given rewritten query $q$ and the original user query $q'$, each agent $a_i$ first submits $q$ to the shopping platform's search interface. The agent then parses the first page of search results, filtering out all sponsored or advertised products, and extracting a list of products $P_q = \{p_1, p_2, ...p_N\}$. For each product $p_j$, it gathers information including the product title, description, image, price, overall rating, the first four customer reviews, and shipping details. For each product $p_j$, the agent $a_i$ assigns a discrete semantic relevance score, $s_{i,j} \in \{-1, 0, 1\}$, corresponding to "Irrelevant", "Partially Relevant", and "Fully Relevant" respectively, along with the reasoning for the given label. We detail the prompt used for the agent in Appendix B.1 and a few examples of scoring done by the agent in Appendix D. After evaluating all products, the agent makes a purchase decision, identifying a subset of products, $P_{buy} \subseteq P_q$, which it would purchase and calculate the total raw purchase value, $p_{raw}$. The prompt for this final purchase is illustrated in Appendix B.2.

**Fitness Function Formualtion** Recent research (Liu et al., 2023b) has demonstrated that utilizing multiple classifiers can accurately predict a fine-grained, continuous numeric score. Therefore, the individual judgments are aggregated into a single, continuous fitness score $F(q)$. First, the semantic score for each product $p_j$ is computed by averaging across all agents: $s_j = \frac{1}{K}\sum_{i=1}^{K} s_{i,j}$. The raw purchase value $p_{raw}$ is normalized to ensure diminishing returns using an exponential transformation: $n = 1 - e^{-\lambda p_{raw}}$, where $\lambda$ is a scaling constant. The final fitness score $F(q)$ is a weighted linear combination of three objectives, reflecting the multi-faceted goals of a real e-commerce platform:

$$F(q) = w_{10} \cdot s_{10} + w_a \cdot s_a + w_p \cdot n$$

where $s_{10}$ is the average semantic score of the top-10 retrieved products, $s_a$ is the average semantic score of all retrieved products, and $n$ is the normalized purchase value. The weights ($w_{10}$, $w_a$, $w_p$) allow for tunable control over the optimization's priorities, such as emphasizing top-of-page relevance versus overall page quality or sales.

## 3.2 Genetic Algorithms for OptAgent

Genetic Algorithms (GAs) are a class of evolutionary algorithms inspired by the process of natural selection. These algorithms serve as optimization and search techniques that emulate the process of natural evolution. The simulation-based fitness function $F(q)$ provides a means to evaluate any given query. We chose a GA as our optimizer due to the unique challenges of optimizing in a subjective domain. Our fitness score from our agent simulation is a helpful but imperfect guide, which introduces stochasticity into the optimization process. Our analysis shows a moderate correlation (Pearson $r = 0.552$) between the agent scores and human judgments, which means the agent scores are a useful but "noisy" estimate of a query's true quality (More details about the human study in Appendix D). A simple search algorithm, such as greedy hill-climbing, could easily get stuck on a local optima. A GA, in contrast, is inherently more robust to a noisy environment. Its population-based search allows it to explore multiple regions of the search space simultaneously, reducing the risk of premature convergence. To discover high-fitness queries, we use GA that iteratively evolves the user query. The overall procedure is detailed in Algorithm 1.

**Initial Population Generation** Initialization policy plays a pivotal role in genetic algorithms because it can significantly influence the algorithm's convergence speed and the quality of the final solution. For OPTAGENT, the initial population $P_0$ is generated by prompting an LLM to create $N$ diverse, semantically similar versions of the user's original query $q_{initial}$. The prompt used for this LLM is shared in Appendix B.3.

**Fitness Evaluation** The fitness $F(q)$ of every query in the current generation's population $P_g$ is computed using the multi-agent simulation described in §3.1. We assign the highest weight to the semantic score of the top-10 products, $w_{10} = 0.5$, followed by the semantic score of all products, $w_a = 0.4$, and finally, on boosting sales, $w_p = 0.1$. This gives us a framework where we make the products highly relevant to the searched query, with a minor objective of boosting sales.

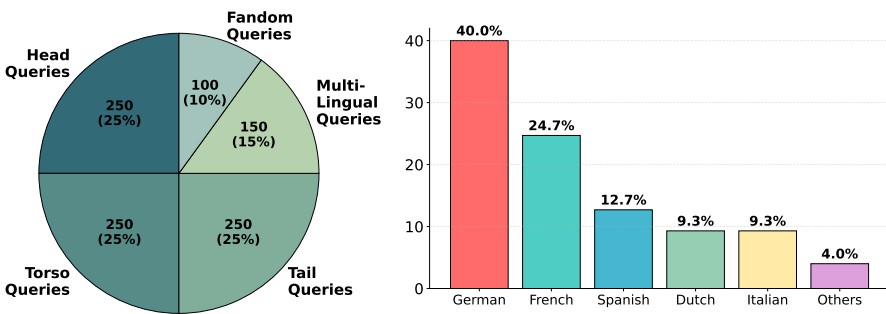

Figure 3: (Left) Data distribution of all the user queries in our dataset. (Right) Distribution of the multi-lingual queries.

**Selection**    We employ an elitism strategy to select candidates for the next generation of queries. The top $M = \alpha N$ queries from $P_g$, ranked by their fitness scores, are directly passed to the next generation, $P_{g+1}$. This ensures that high-performing solutions are preserved in the next generation.

**Crossover**    To fill the remaining $N - M$ slots in the new population, two parent queries are selected from the previous generation. With probability $p_{crossover}$, they are passed to an LLM prompted to perform a "crossover" operation, i.e., creating a new child query that intelligently combines meaningful semantic elements from both parents. We illustrate the prompt in Appendix B.4.

**Mutation**    With probability $p_{mutation}$, a selected query undergoes "mutation". It is passed to an LLM with a prompt that instructs it to make a small but meaningful alteration (e.g., using a synonym, reordering words) to create a new variant. We detail the prompt in Appendix B.5.

The above process repeats for a fixed number of generations, $G$, or until the coverage criterion is met. The final output is the query with the highest fitness score found throughout the entire evolutionary process.

## 4    EXPERIMENT SETUP

### 4.1    DATASET

We collected a dataset of 1000 real-world queries submitted by users on the Etsy e-shopping website[2]. Each query was manually reviewed to ensure that no Personally Identifiable Information (PII) was present. The dataset is categorized into five distinct classes, as shown in Figure 3, to allow for fine-grained analysis. *Head Queries* are high-frequency, popular search terms. They constitute the top 4% of queries in terms of popularity. *Torso Queries* are moderately frequent search terms between 70% and 96% in terms of popularity, while *Tail Queries* are infrequent queries that constitute less than 70% in terms of popularity. *Fandom Queries* are queries related to a particular franchise (e.g., TV Show or Movies). *Multi-Lingual Queries* are queries in non-English languages. Out of 150 multi-lingual queries, a majority (60) are in German, followed by French (37). Languages like Hmong, Turkish, Hindi, or Romanian constitute the "Others" part of multi-lingual queries. Popular queries tend to contain fewer mistakes and are more friendly to the recommendation systems; however, a majority of the queries that are searched are unique and infrequent, which make up Torso and Tail sub-sections of our dataset.

### 4.2    MODELS AND METRICS

We use Gemini-2.5-Flash (Comanici et al., 2025) and run OPTAGENT with the following specifications: $\lambda = 0.02$, $w_p = 0.1$, $w_{10} = 0.5$, $w_a = 0.4$, $N = 5$, $G = 4$, $\alpha = 0.6$, $p_{crossover} = 0.7$, and $p_{mutation} = 0.1$, as defined in §3. We report the total cost of OPTAGENT in Appendix E. For our evaluation agent, we use $K = 5$ agents with temperatures $\{0.00, 0.25, 0.50, 0.75, 1.00\}$.

---

[2]https://www.etsy.com

We use $F(q)$ defined in §3.1 as our metric for evaluation. To validate the reliability of our agentic evaluation, we randomly sampled 287 queries and had them annotated by human experts for semantic relevance. Since, average of semantic scores results in a continuous score, we report the Pearson correlation between human annotators and our evaluation agent. Our evaluation agent shows a statistically significant positive correlation with human annotations (Pearson $r = 0.552$, $p < 0.001$), indicating a moderate and meaningful alignment with human judgment. Notably, due to the subjective nature of our task, agreement between independent human annotations is also moderate alignment (Pearson $r = 0.532$, $p < 0.001$). Full details of annotation are described in Appendix D.

### 4.3 BASELINES

We compare OPTAGENTagainst three baselines: 1) User Query: The original, unmodified query submitted by the user. This serves as our control baseline. 2) LLM-Rewrite: A simple and standard approach where an LLM is prompted to rewrite the user query. 3) Best-of-N (BoN) Rewrite: This is an inference-time technique (Snell et al., 2024), where we prompt an LLM to generate 8 (same average number of queries as OPTAGENT) different candidate rewrites for the query and then use our multi-agent simulation to evaluate all candidates. The one with the highest score is selected.

## 5 RESULTS

### 5.1 PERFORMANCE OF OPTAGENT

Table 1: Mean fitness scores of all methods across different query subsections. OPTAGENT achieves the highest score in every category, demonstrating robust performance. The percentage improvement of OPTAGENT over the BoN-Rewrite is shown in parentheses.

| Query Subsection | Method | | | |
|---|---|---|---|---|
| | User Query | LLM-Rewrite | BoN-Rewrite | OPTAGENT |
| Head Queries | 0.6433 | 0.5744 | 0.7493 | **0.7660** (2.23% ↑) |
| Torso Queries | 0.5741 | 0.5335 | 0.7106 | **0.7377** (3.81% ↑) |
| Tail Queries | 0.4978 | 0.4841 | 0.6129 | **0.6405** (4.50% ↑) |
| Fandom Queries | 0.7328 | 0.6335 | 0.7762 | **0.7969** (2.67% ↑) |
| Multi-Lingual Queries | 0.7189 | 0.6833 | 0.8260 | **0.8547** (3.47% ↑) |
| All Queries | 0.6100 | 0.5509 | 0.7199 | **0.7441** (3.36% ↑) |

Table 1 presents the main results across all categories. OPTAGENT consistently achieves the highest scores, outperforming all baselines. On average, OPTAGENT improves the query fitness by 21.98% over the original User Query and 3.36% over the BoN-Rewriting.

An interesting trend is that the naive LLM-Rewrite often performs worse than simply using the original user query. This highlights the non-trivial nature of the QR task, where a single, unguided rewrite attempt by an LLM can easily misinterpret intent or generate a semantically correct but less effective query. The substantial improvement of BoN-Rewrite over both User Query (+18.02%) and LLM-Rewrite demonstrates the critical importance of generating multiple candidates and having a reliable evaluation mechanism to select the best one. OPTAGENT builds upon this, showing that a guided evolutionary search can explore the solution space more effectively than simple sampling, yielding further performance gains.

**Performance across Query Subsections** The performance of OPTAGENT varies across different query types. The largest relative improvement from the original query is observed in *Tail Queries* (28.67%), followed by *Torso Queries* (28.50%) and *Head Queries* (19.07%). Tail queries are significantly difficult to optimize using traditional methods because they lack the historical data needed to learn rewrite patterns. Our result suggests that the exploratory nature of evolutionary search is especially effective in these data-sparse, high-uncertainty scenarios. Conversely, *Fandom Queries* see the smallest improvement (8.74%), likely because users searching for specific franchise-related items are already quite precise, leaving less room for optimization.

We observe an average improvement of $18.89\%$ on *Multi-Lingual Queries* over the user query. Table 2 shows the breakdown for all languages. OPTAGENTprovides consistent improvements across all languages, with the largest gain in Italian ($32.36\%$). The "Others" category, which includes lower-resource languages, shows the smallest gain and a negligible improvement over the BoN baseline ($0.49\%$). This aligns with prior research (Huang et al., 2024), indicating that LLMs exhibit weaker reasoning capabilities in low-resource languages, suggesting that both the generative and evaluative capacities of the agents are less effective in these cases.

Table 2: Mean fitness scores for multi-lingual queries. OPTAGENT consistently outperforms baselines across different languages. The percentage improvement of OPTAGENT over BoN-Rewrite is shown in parentheses.

| Language | Method | | | |
|---|---|---|---|---|
| | User Query | LLM-Rewrite | BoN-Rewrite | OPTAGENT |
| German | 0.7107 | 0.6331 | 0.8139 | **0.8432** (3.59% ↑) |
| French | 0.7343 | 0.6759 | 0.8464 | **0.8721** (3.03% ↑) |
| Spanish | 0.7621 | 0.7299 | 0.8780 | **0.9012** (2.65% ↑) |
| Dutch | 0.7755 | 0.6837 | 0.8411 | **0.8692** (3.34% ↑) |
| Italian | 0.6417 | 0.7142 | 0.7960 | **0.8493** (6.70% ↑) |
| Others | 0.6167 | 0.5161 | 0.6901 | **0.6931** (0.49% ↑) |

**Performance across Generations**

Figure 4 illustrates the average fitness of the best query in the population across the four generations of the evolutionary process. We observe a consistent increase in fitness with each generation, though the rate of improvement diminishes over time, suggesting that the algorithm is converging towards a good solution. This demonstrates that the evolutionary operators of crossover and mutation are effectively discovering better queries over time. An analysis of the fitness components reveals that in the first generation, $97.26\%$ of the fitness gain comes from improving the semantic relevance scores. By the final generation, the contribution from the normalized purchase value increases to $5.87\%$, indicating that the algo-

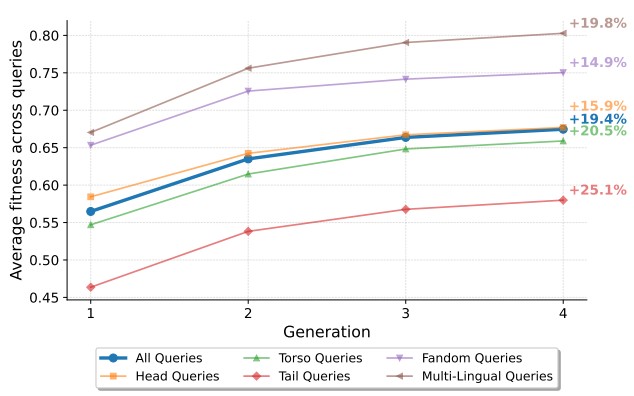

Figure 4: Average fitness of the best query in the population across four generations for each query subsection. Fitness consistently improves, with diminishing results in the later generations.

rithm first prioritizes finding relevant products before fine-tuning to encourage purchasing behavior. Interestingly, in the case of *Fandom Queries*, we observe these metrics directly competing against each other as the purchase value actually decreases for two generations, making an average of $0.2\%$ decrease and then increasing along with semantic scores in the final generation.

## 5.2 ABLATION OF OPTAGENT

To better understand the contribution of each component of OPTAGENT, we conduct an ablation study, the results of which are presented in Table 3. The full OPTAGENT framework serves as our point of comparison.

Removing the evolutionary operators entirely (which is equivalent to the BoN-Rewrite baseline with 5 different rephrases) results in the largest performance drop of $6.4\%$, confirming that the guided search of the genetic algorithm is the most critical component for achieving peak performance. Further ablation reveals that a major portion of this drop is contributed by the crossover operation ($6.1\%$), while mutation contributes ($0.3\%$). While both these operators play a major role, crossover

plays a much bigger role in OPTAGENT compared to mutation, with $83.4\%$ of queries observed to have lower scores without crossover. We hypothesize that the reduced impact of the mutation operation is because LLMs, by themselves, are unable to rephrase queries in a meaningful way. This also explains why BoN-Rewrite baseline underperforms when compared to OPTAGENT.

Table 3: Ablation study of OPTAGENT components on the full dataset. Each row shows the performance when a specific component is removed, highlighting its contribution to the final result.

| Method | Overall Fitness Score | $\Delta$ vs. OPTAGENT |
|---|---|---|
| **OPTAGENT (Full Framework)** | **0.7441** | - |
| - Evolutionary Operations (i.e., BoN-Rewrite) | 0.6965 | $6.4\%\downarrow$ |
| - Crossover | 0.6987 | $6.1\%\downarrow$ |
| - Mutation | 0.7419 | $0.3\%\downarrow$ |

## 5.3 EVALUATION AGENT ANALYSIS

We observe a Moderate Agreement between human annotators and our evaluation agents. We detail the full human annotation in Appendix D and detail some examples in Appendix C. We observe that, in most cases, our agent makes reasonable deductions; however, there exist two failure cases. First, for some products, some information is hidden in interactive functionalities (e.g., in dropdown menus) of the website, which our agent fails to parse. This is especially relevant in queries where the user specifies any quality of a product, e.g., color or size. Second, we observe instances where our agent relies too heavily on customer reviews (or a lack thereof) when rating the product. This is especially relevant for newer products that have limited or no reviews, even if the product itself is relevant to the user's query.

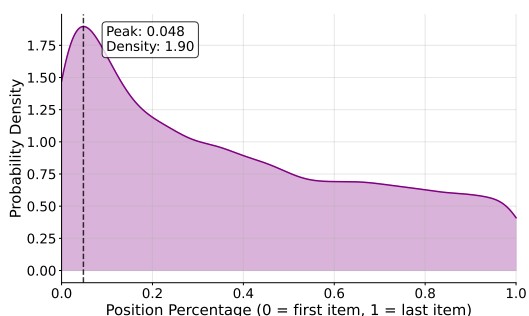

Figure 5: Probability Distribution of our evaluation agent in selecting products for purchase out of all listed products. Similar to real users, our evaluation agent highly prefers products listed in the beginning compared to products later down the search results.

Finally, we study how our evaluator agent selects products for purchase from all the products listed on the first page of search results. Fig. 5 represents the probability distribution of our agent in selecting a product from all listed products. We observe that the agent prefers to purchase the first few products with a much higher probability compared to the last few. Although this is an observed bias, this behavior is very similar to how real users behave on real e-commerce platforms (Wang et al., 2023; Collins et al., 2018). Therefore, we believe this makes our evaluation agent more reliable for simulating real user behavior on e-commerce platforms.

## 6 CONCLUSION

In this work, we addressed the fundamental challenge of optimizing LLMs in subjective domains where traditional reward signals are unavailable. We introduced OPTAGENT, a novel framework for e-commerce query rewriting, that replaces static reward functions with a dynamic fitness evaluation derived from a multi-agent simulation. By using an ensemble of LLM agents with diverse reasoning paths, our evaluation creates a rich, nuanced fitness landscape that better captures the complexity of latent human preference. When coupled with an LLM-powered genetic algorithm, our approach significantly outperforms strong baselines, particularly for difficult, long-tail queries. Our work provides a generalizable and scalable blueprint for optimization in the absence of explicit rewards, opening new avenues for developing more capable and aligned AI systems in a wide range of human-centric applications.

REPRODUBILITY STATEMENT

To ensure the reproducibility of our results, we detail OPTAGENT in algorithm 1 and detail the prompts in Appendix B. We commit to releasing the following upon publication: (1) The complete source code for the OPTAGENT framework, including the implementation of the evolutionary algorithm and the multi-agent simulation. (2) The full set of prompts and agent tools used by the evaluation agents and the genetic operators (crossover and mutation). (3) The dataset of the e-commerce queries used in our experiments. All experiments were conducted using the Gemini-2.5-Flash model. The key hyperparameters for the evolutionary algorithm are detailed in §4.2.

ETHICS STATEMENT

The query dataset used in this study was sourced from real user data. We undertook a rigorous manual review process to identify and remove any Personally Identifiable Information (PII) to protect user privacy. We acknowledge that the LLM agents used in our simulation may inherit biases present in their training data. Our use of a multi-agent ensemble with diverse temperatures is a deliberate design choice intended to mitigate the impact of any single agent's bias by aggregating multiple, varied perspectives. However, the potential for correlated biases across the ensemble remains an area for future investigation. Furthermore, in accordance with ICLR policy, we disclose that AI assistants, specifically Grammarly for grammar correction and ChatGPT for sentence restructuring and paraphrasing, were utilized during the preparation of this manuscript. The authors have reviewed, edited, and take full responsibility for all final content presented in this paper.

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

## A  OPTAGENT ALGORITHM

Algorithm 1 details the overall algorithm for OPTAGENT.

---

**Algorithm 1** OPTAGENT

---

1: **procedure** OPTAGENT($q_{\text{initial}}, N, G, \alpha, p_{crossover}, p_{mutation}$)
2:    $P_0 \leftarrow$ INITIALIZE_POPULATION($q_{\text{initial}}, N$)        ▷ Initial Population Generation
3:    **for** $g = 0$ to $G - 1$ **do**        ▷ Evolutionary Loop
4:        **for** each $q \in P_g$ **do**
5:            $F(q) \leftarrow$ AGENTIC_FITNESS_SIMULATION($q$)    ▷ Fitness Evaluation
6:        **end for**
7:        $P_{g+1} \leftarrow$ Top $\alpha N$ queries from $P_g$ sorted by $F(q)$    ▷ Selection (Elitism)
8:        **while** $|P_{g+1}| < N$ **do**    ▷ Next Population Generation
9:            $q_{\text{parent1}} \leftarrow$ SELECT($P_g$)
10:            **if** rand() $< p_{crossover}$ **then**
11:                $q_{\text{parent2}} \leftarrow$ SELECT($P_g$)
12:                $q_{\text{child}} \leftarrow$ CROSSOVER($q_{\text{parent1}}, q_{\text{parent2}}$)
13:            **else**
14:                $q_{\text{child}} \leftarrow q_{\text{parent1}}$
15:            **end if**
16:            **if** rand() $< p_{mutation}$ **then**
17:                $q_{\text{child}} \leftarrow$ MUTATE($q_{\text{child}}$)
18:            **end if**
19:            Add $q_{\text{child}}$ to $P_{g+1}$
20:        **end while**
21:    **end for**
22:    **return** best query from $P_G$
23: **end procedure**

---

## B  PROMPTS USED BY OPTAGENT

### B.1  EVALUATION ANALYSIS PROMPT

You are a product analyst for Etsy, an online shopping platform. Based on the provided product image, the searched query, product price, seller information, any available customer reviews, and shipping/delivery information, give me a detailed analysis of the product. Analyze the product using analytical thinking and common sense to determine its semantic relevance to the searched query.

When product price is provided, consider the value proposition and the perceived value based on quality and features.

When seller information is provided, consider the seller's reputation and relevance to the searched query. Some searches may be looking for products from specific sellers or brands.

When customer reviews are provided, use them to gain insights into product quality, user satisfaction, potential issues, and real-world usage experiences. Consider how the reviews support or contradict your visual analysis.

When shipping and delivery information is provided, factor in delivery times, shipping costs, and availability in your analysis. Consider how these logistics aspects might affect the purchasing decision.

Please provide your analysis in a JSON format with the following keys:

- "summary": A comprehensive summary of your overall analysis, including both positive and negative aspects of the product in relation to the searched query.
- "semantic_score": Select either "HIGHLY RELEVANT", "SOMEWHAT RELEVANT", or "NOT RELEVANT" based on how well the product matches the searched query.

**EXAMPLE INPUT 1**

Searched Query: healthy energy drink
Current Date: May 21

**OUTPUT 1**
{
"summary": "This product has several positive aspects: it's affordable at $1.00, the seller has a high rating with 1,470 reviews, and it can be delivered on May 22 (1 day delivery). However, the critical issue is that the product is a healthy snack, not a drink, which completely misses the specific search for a healthy energy drink.",
"semantic_score": "NOT RELEVANT"
}

**EXAMPLE INPUT 2**

Searched Query: House of Staunton Chess Set
Current Date: June 13

**OUTPUT 2**
{
"summary": "The product is a high-quality wooden chess set that can be personalized, with reasonable delivery time (June 15, 2 days from now). On the positive side, it's made of quality wood and offers customization. However, there are several drawbacks: only 3 total reviews providing limited social proof, the price is somewhat high for a chess set, and most importantly, it is not from the House of Staunton brand as specifically searched for. While it is a chess set, it doesn't match the brand requirement.", "semantic_score": "SOMEWHAT RELEVANT"
}

**EXAMPLE INPUT 3**

Searched Query: renaissance-style necklace
Current Date: September 10

**OUTPUT 3**
{
"summary": "This is a high-quality renaissance-style necklace with excellent reviews (500 reviews with high rating) and craftsmanship that perfectly matches the searched query aesthetic. The only drawback is the shipping date of September 20, which is more than a week away. Despite the longer shipping time, the product strongly aligns with the searched renaissance-style necklace criteria.",
"semantic_score": "HIGHLY RELEVANT"
}

## B.2 FINAL PURCHASE PROMPT

You are making a purchase decision based on the given query and the products. Analyze the products and make a purchase decision based on analytical thinking and common sense.

Inputs you will receive:
- The searched query.
- A list of products and their price and a short summary.

Your job:
1. Critically compare the products based on their relevance to the searched query.
2. Decide which product(s) (one or more) should be purchased based on your reasoning.
3. Buy a reasonable number of products, depending on the price and the query. Act like a real customer.
4. Provide a short, logical justification followed by the list of product names that you want to purchase.
5. If none of the products should be purchased, explain why and return an empty list of recommendations.

Return ONLY a valid JSON object with the following structure. Do not include any other text or comments.

**OUTPUT STRUCTURE**
{
"reasoning": "<explanation of why you chose or rejected the products>",
"recommendations": [
"<product_name_1>",
"<product_name_2>",
...
]
}

## B.3 INITIAL POPULATION GENERATION PROMPT

You are an expert at creating variations of shopping queries for e-commerce platforms like Etsy. Given an original query, generate semantically similar but diverse variations that could potentially find better or different relevant products.

Guidelines:
- Keep variations relevant to the original intent
- Use synonyms, related terms, and different phrasings
- Consider different product attributes (size, color, style, material)
- Include both broader and more specific versions
- Each variation should be a single line, natural search query
- Variations should be 2-8 words typically
- Return the variations as a JSON list of strings.

## B.4 CROSSOVER PROMPT

You are an expert at combining shopping queries to create new, potentially better variations. Given two parent queries, create a new query that combines the best aspects of both while maintaining search relevance.

Guidelines:
- The result should be a natural, searchable query
- Combine meaningful elements from both parents
- Keep it concise (2-8 words typically)
- Maintain the original search intent
- Just return the new query, no other text.

## B.5 MUTATION PROMPT

> You are an expert at creating subtle variations of shopping queries while maintaining their core meaning and search intent. You will be given a query and a summarized feedback for this query. Return the revised query, addressing the feedback.
>
> Guidelines:
> - Keep the same general length (don't make it significantly longer or shorter)
> - Maintain the original search intent
> - Make small but meaningful changes (synonyms, reordering, slight modifications)
> - The result should still be a natural, searchable query
> - Just return the revised query, no other text.
> - Try to address the feedback in the revised query.

## C EXAMPLES

### C.1 CASE STUDY OF BIAS IN PERSONA FOR ANALYSIS AGENT

One key problem while using an ensemble of agents is wasted compute if several agents give the same reasoning. To mitigate this, several works have used personas to add personality to these agents. While effective, this can introduce undesirable behaviors. We use the personas from (Lu et al., 2025) and detail one such instance of bias below:

> **Benign Persona**
>
> Persona: Michael
>
> Background:
> Michael is a 41-year-old man who has built a successful career in the tech industry, leveraging his analytical skills and entrepreneurial spirit. Despite his professional accomplishments, he remains grounded and values work-life balance, making time for his hobbies and charitable endeavors.
>
> Demographics:
> Age: 41
> Gender: Male
> Education: Bachelor's Degree in Computer Science
> Profession: Software Engineer, Tech Startup Co-Founder
> Income: $150,000
>
> Financial Situation:
> As a software engineer and tech startup co-founder, Michael's income falls within the $94,001 to $1,000,000 range. He has a strong financial footing, with a healthy savings account and investments that provide a sense of financial security. While he is mindful of his spending, he also enjoys the occasional splurge on high-quality items that align with his personal values and interests.
>
> Shopping Habits:
> Michael's shopping habits reflect his practical yet discerning approach to consumption. He tends to invest in durable, well-crafted products that will stand the test of time, whether it's a high-quality suit for work or a state-of-the-art piece of home electronics. He is also conscious of his environmental impact and seeks out sustainable and ethical brands when possible.
>
> Professional Life:

Michael's professional life is a source of both fulfillment and inspiration. As a software engineer, he is constantly learning and pushing the boundaries of his field, working on innovative projects that solve real-world problems. His involvement in a tech startup allows him to exercise his entrepreneurial spirit, collaborating with a team to bring their vision to life.

Personal Style:
Michael's personal style is understated yet refined. He favors a classic, minimalist aesthetic, with a wardrobe that consists of well-fitting, high-quality clothing in neutral tones. He values functionality and comfort, but also appreciates the importance of presenting a polished and professional appearance. In his free time, he enjoys exploring his creative side through hobbies like photography and woodworking.

Personal Life:
Outside of his professional pursuits, Michael is an avid outdoorsman, finding solace in activities like hiking, camping, and mountain biking. He is also deeply invested in his community, serving on the board of a local non-profit organization that provides educational resources to underprivileged youth. Michael understands the importance of work-life balance and makes a conscious effort to maintain strong relationships with his family and close friends.

---

**Racial Bias observed in Evaluation Agent when analyzing a product**

The product is a direct match for Michael's specific search query and is affordably priced with fast, free shipping and a return policy. **However, its origin from China might not fully align with Michael's discerning shopping habits, which prioritize ethical sourcing, high quality, and robust product research**. The nature of an 'action figure' also might not perfectly fit his preference for 'durable, well-crafted' items that stand the test of time, unless this is a new or casual interest.

---

In the above example, since the product is made in China, the LLM incorrectly assumes it is lower quality or sourced unethically. Since proprietary LLMs are trained to be unbiased, we chose not to assign a persona to our agent; instead, we rely on varying temperatures to generate different reasoning paths.

## C.2 EVALUATION AGENT EXAMPLES

Below, we detail two instances where our evaluation agent fails:

**Interactive Elements** For the query "bridesmaid green wine tumbler", the product description or image doesn't highlight green, but is present in the dropdown menu [3], making it a **Fully Relevant** product. The following is the agent's reasoning:

---

This product is a highly-rated 'wine tumbler' with excellent customer satisfaction, evidenced by its 4.8 out of 5-star rating from 917 reviews. Customers praise its overall quality as bridesmaid gifts. The price of $11.90 is reasonable for such an item, and the estimated delivery time (10-12 days) is standard. However, the critical issue is that the searched query 'bridesmaid green wine tumbler' strongly implies a search for the color green. There is no indication that it is available in the color green. While it is a high-quality wine tumbler, it does not fulfill the color-specific nature of the search query.

Semantic Classification: **Partially Relevant**

---

[3]https://www.etsy.com/listing/1877369188

**Heavy Reliance on Customer Reviews**  For the query "freida mcfadden", the product [4] has no reviews but is a **Fully Relevant** product for the query. Following is the reasoning and classification given by the agent:

> This product explicitly features 'Freida McFadden' in its name and displays numerous book covers by the author in the image, making it highly relevant to the searched query for the author. It is very affordable at \$3.50 for a 'collection' and is available via instant download, which are positive aspects. However, the critical ambiguity lies in the fact that no customer reviews are provided to clarify the quality of the product, and the generic seller name 'KraftyBStudio' doesn't offer specific credibility for providing official e-book content.
>
> Semantic Classification: **Partially Relevant**

## D  HUMAN ANNOTATION

To validate the effectiveness of our agents acting as judges as a proxy for human judgment, we conducted a human annotation study. This section details the protocol used for collecting and analyzing human relevance scores.

We recruited multiple graduate students and e-commerce experts for annotation. Annotators were presented with the query and a corresponding product retrieved by our agent. Annotators could visit the product on the website to see a more detailed description. The annotator's task was to evaluate the semantic relevance of the product given the query.

Annotators were asked to assign a single relevance score on a 3-point Likert scale, based on the following rubric:

1. Fully Relevant: The product accurately captures the original query, including the user's intent.

2. Partially Relevant: The product captures some aspects of the query but may be too broad, too narrow, or slightly off-topic.

3. Irrelevant: The product significantly misinterprets the query intent and has no connection to the query. If you see this product among the top search results of the query, you will think the search function is broken.

A random sample of 287 (query, product) pairs was selected from our test set, stratified across the different methods and query categories to ensure a representative sample. Each of the 287 pairs was evaluated by two independent annotators. We measured the inter-annotator agreement using Quadratic Cohen Kappa, a standard metric for reliability with multiple raters. The resulting agreement was $\kappa = 0.5392$, which indicates "Moderate Agreement" and a $64.1\%$ exact match. We convert the classification labels into $+1$, $0$, and $-1$ respectively, and the final human score for each query was taken as the average of the two annotators' scores.

As reported in the main text (§4.2), we calculated the Pearson correlation coefficient between our agent-derived fitness scores ($F(q)$) and the average human relevance scores. The analysis yielded a statistically significant positive correlation of $r = 0.552$ ($p < 0.001$), validating that our agentic simulation serves as a meaningful and moderately strong proxy for human preference in this subjective task.

## E  COST OF OPTAGENT

To provide transparency regarding the resources required to run our framework, this section details the computational cost of optimizing a single query using OPTAGENT. The costs are based on the experimental setup described in Section 4.2. The total cost is a function of the number of LLM API calls and the number of tokens processed in each call. Our process for a single query involves:

---

[4] https://www.etsy.com/listing/4332712379

1. **Initialization:** 1 LLM call to generate the initial population of 5 queries.

2. **Evolutionary Loop (4 Generations):**

   (a) **Fitness Evaluation:** For each of the 5 queries in a generation's population, 5 agents perform an evaluation. This involves each agent making approximately 60 calls to score individual products on a search results page, followed by 1 call to make a final purchase decision. This results in 5 queries $\times$ 5 agents $\times$ (60 product scores + 1 purchase decision) = 1525 LLM calls per generation.

   (b) **Genetic Operators:** To create the next generation, an average of 2 LLM calls are made for crossover and mutation operations.

This results in a total of approximately $1 + 4 * (1525 + 2) = 6109$ LLM calls to optimize a single query, with the majority of calls coming from evaluating the quality of the query rather than optimizing it.

We estimate the token usage based on the prompts detailed in Appendix B and typical product page content. The pricing for Gemini-2.5-Flash is \$0.30 per 1 million input tokens and \$2.50 per 1 million output tokens at the time of our experiments [5]. Our **Estimated Total Cost per Query: \$8**, making the total cost of OPTAGENT on a set of 1000 queries to be $\sim \$8k$

---

[5]Current Pricing: `https://ai.google.dev/gemini-api/docs/pricing`

