# OpenReview forum: "OptAgent: Optimizing Query Rewriting for E-commerce via Multi-Agent Simulation"
_ICLR.cc/2026/Conference — ICLR 2026 Conference Withdrawn Submission_

### Official Review · Reviewer_uWUL · 2025-10-29

**Soundness:** 2
**Presentation:** 3
**Contribution:** 2
**Rating:** 2
**Confidence:** 2

**Summary:**

This paper addresses the challenge of e-commerce query rewriting (QR): user inputs are often ambiguous, and evaluation is subjective with no clear standards. It uses multiple LLM agents to simulate customers (enhancing diversity through temperature sampling) to generate dynamic reward signals, combined with a genetic algorithm (crossover and mutation in natural language) to iteratively optimize queries without relying on historical data, particularly helping tail queries capture user intent accurately.

Key contributions include:

- A multi-agent simulation serving as a robust fitness function for evaluation;

- The OptAgent framework, which replaces static rewards with dynamic scoring;

- On 1,000 Etsy queries, relevance improved by 21.98% over the original queries and 3.36% over the BoN baseline (4.5% for tail queries).

**Strengths:**

The paper's key strength lies in its multi-agent simulation for query rewriting evaluation. Using LLM agents with varied temperatures (0.00–1.00), it creates dynamic reward signals that better approximate human preferences, reducing biases and enabling robust fitness scores through aggregated semantic relevance and purchase decisions.

Another strength is its genetic algorithm optimization, which refines queries via natural language crossover and mutation. This population-based approach enhances exploration in subjective domains.

**Weaknesses:**

(1) The multi-agent simulation uses isolated agents that only differ in temperature, lacking interaction which could better mimic real shopper dynamics. Please explain the reasons for not using multi-agent interaction.

(2) The inference-time cost is not adequately discussed. The high computational demand of ~1,525 LLM calls per generation casts doubt on the method's feasibility for high-volume use.

(3) Comparisons are limited to basic baselines like BoN,  and the reliance on a single LLM (Gemini) for both optimization and evaluation raises concerns about the generalizability of the findings.

**Questions:**

Please refer to the above.

---

### Official Review · Reviewer_hd5b · 2025-10-31

**Soundness:** 2
**Presentation:** 2
**Contribution:** 2
**Rating:** 4
**Confidence:** 2

**Summary:**

This paper introduces a framework called optagent to optimize E-commerce query rewriting. It is a multi-agent framework that applies genetic algorithms. In a dataset of 1000 queries, it achieves better performance than baselines.

**Strengths:**

1. This paper applies the genetic algorithm, which is an interesting approach.
2. It uses multiple agents to simulate user behaviors.
3. This paper focuses on query rewriting in e-commerce, which can benefit both customers and sellers.

**Weaknesses:**

1. The introduction cites information from unofficial websites. It is recommended to valid these sources with official documents such as financial statements to enhance credibility.
2. The repository link is not provided, which hinders reproducibility.
3. The dataset is relatively small (only 1,000 queries), which may limit the generalizability of the results. Consider expanding the dataset or using other datasets.
4. The problem of efficiency. Running multiple LLM agents over multiple generations is expensive. The paper defers cost details to the appendix, but comparative analysis on compute vs. gain tradeoff is lacking.

**Questions:**

1. For a query, what is the average efficiency of your method?
2. Do you consider adding an RL-based method for query rewriting as a baseline?

---

### Official Review · Reviewer_4hAu · 2025-11-01

**Soundness:** 3
**Presentation:** 3
**Contribution:** 2
**Rating:** 2
**Confidence:** 4

**Summary:**

The paper proposes OPTAGENT, which couples a multi-LLM “shopper” simulation with a genetic algorithm to optimize e-commerce query rewriting. The ensemble’s averaged scores serve as the fitness for evolution; reported gains are +21.98% over original queries and +3.36% over a Best-of-N LLM baseline.

**Strengths:**

1. Clear problem framing for subjective, label-scarce QR; replaces a single judge with a simulated crowd.
2. Simple, reusable mechanics: temperature-diverse evaluators; crossover/mutation directly in natural language.

**Weaknesses:**

1. Evaluator–optimizer coupling. Training and testing both rely on the same agentic evaluator; human correlation is only moderate (Pearson r=0.552), so overfitting to the evaluator remains likely. Needs independent metrics or online A/B evidence.
2. Baselines are weak. Comparisons cover user query, single LLM rewrite, and BoN only; missing stronger QR baselines common in practice.
3. External bias not controlled. Authors show strong position bias in purchases toward early-ranked items (Fig. 5) yet do not correct for it in fitness or evaluation (e.g., randomization, PBM/DBN).
4. Reproducibility and generalization risk. Dataset is from a single platform (Etsy). Results depend on live search pages and site behavior; claims of code release do not mitigate the moving-target evaluation environment. No evidence on other domains.

**Questions:**

1. Baselines and Generalization: The paper mentions "LLM-as-a-Judge" similar to RLAIF and criticizes single LLM judges for biases, while noting Zuo et al.'s heavy reliance on "vast amounts of historical user interaction data." Beyond BoN, why not compare with RLAIF or session-graph-based QR methods? How does the framework generalize to other subjective tasks (e.g., personalization in recommendation systems)? What are the details of translation handling for multilingual queries (e.g., using LLM translation)?
2. Human Evaluation Details: What are the sample size, annotator diversity (e.g., cultural backgrounds) in the human study in Appendix D? Are the agent bias cases (Appendix C.1) mitigated during the optimization process? If budget allows, is there a plan for A/B testing on real Etsy data?

---

### Official Review · Reviewer_jWo7 · 2025-11-02

**Soundness:** 2
**Presentation:** 3
**Contribution:** 2
**Rating:** 4
**Confidence:** 4

**Summary:**

- The paper introduces OptAgent, a framework designed to optimize e-commerce query rewriting through the use of multi-agent simulations and genetic algorithms.
- OptAgent employs a population of LLM-based agents to simulate user evaluations by assessing the relevance of retrieved products. The ensemble's average rating functions as a fitness score for an evolutionary algorithm, which incrementally refines the user’s query using crossover and mutation operators executed by LLMs.
- Experiments conducted on a newly developed dataset demonstrate significant improvements over all baselines. Comprehensive analyses of agent behaviors, task categories, and ablation studies on OptAgent components enhance the paper's completeness.

**Strengths:**

- Query rewriting for e-commerce is a practical and underexplored setting for multi-agent systems. The framing of "multi-agent simulation + evolutionary algorithm" is creative and relevant.
- The paper introduces a curated dataset of real-world e-commerce queries, which can support future research in this area.
- The paper is well-written, with intuitive figures and detailed algorithmic breakdowns, making the method easy to follow.

**Weaknesses:**

- The proposed evaluation function does not directly measure how well the rewritten query satisfies the user’s latent intent. Instead, it measures product–query relevance as judged by LLMs. This proxy may not reflect the actual goal of e-commerce query rewriting, which is user satisfaction or purchase fulfillment.
- The same evaluation metric (multi-agent fitness) is used for both optimization and final evaluation, which is conceptually similar to evaluating on the training signal. This may inflate reported improvements and is not a standard practice in ML evaluation.
- The baselines (LLM-Rewrite, BoN-Rewrite) are relatively weak. The absence of comparisons to stronger reasoning or optimization approaches (e.g., CoT-SC, MAD, or RL-based fine-tuning methods like SFT or RLVR) makes it hard to gauge the true competitiveness of OptAgent.
- The reported metric improvements are entirely simulation-based. There is no direct user or behavioral validation (e.g., click or purchase data). The link between simulated multi-agent evaluation and real-world e-commerce outcomes remains unverified.

**Questions:**

- The current evaluation emphasizes product relevance but does not explicitly account for users’ latent intent. How can you ensure that the multi-agent scoring meaningfully reflects actual user satisfaction? A more robust evaluation might include fine-grained annotation of latent intent or user studies involving real participants.
- Employing the same fitness function for both optimization and evaluation introduces a risk of overfitting. Could you justify this strategy by referencing previous works that have successfully adopted a similar approach, or alternatively, incorporate an independent evaluation metric, such as human assessments or retrieval-based evaluations, to better demonstrate generalization?
- In some cases, the LLM baseline underperforms compared to the original query. Could you provide qualitative analysis or illustrative examples highlighting the scenarios where direct rewriting results in degraded performance, especially considering that an LLM could theoretically opt to retain the original query if it is already optimal?
- To more thoroughly demonstrate OptAgent’s strengths, it would be beneficial to compare with stronger baselines such as CoT-SC and MAD. Additionally, incorporating baselines that use trained models, such as SFT or RLVR on smaller model sizes, would further reinforce the rigor of your evaluation.
- The current approach ensembles multiple agents by varying the sampling temperature, rather than using different underlying models or roles. As this is not a conventional ensembling strategy, it would be helpful to include an ablation study or further analysis to motivate and support this design choice.

---

### Official Review · Reviewer_J8mF · 2025-11-02

**Soundness:** 2
**Presentation:** 3
**Contribution:** 2
**Rating:** 4
**Confidence:** 3

**Summary:**

The paper introduces OptAgent, a multi-agent framework for e-commerce query rewriting that (i) aggregates judgments from diverse LLM agents into a modular fitness signal and (ii) applies genetic search to optimize candidate rewrites at test time. On an Etsy query set spanning head/torso/tail, fandom, and multilingual segments, OptAgent outperforms the original user queries and Best-of-N baselines.

**Strengths:**

1. The paper introduces a practical and empirically effective approach that couples a multi-agent, policy-guided evaluator with genetic search over query rewrites, delivering clear gains. Incorporating genetic search into the multi-agent optimization workflow offers a measure of novelty.

2. The paper offers a comprehensive evaluation spanning multiple query segments (head/torso/tail, fandom, multilingual).

**Weaknesses:**

1. Using the same multi-agent system to both grade and guide the search can make offline results look better than they really are, because the optimizer may learn the judge’s preference instead of what real users want; if you swap in a different judge (e.g., another model or prompt), the gains may shrink, suggesting overfitting to the original judge.

2. The baselines could be stronger, because gains over the user’s original query and a Best-of-N sampler alone don’t establish progress; beyond BoN, the paper should include baselines from more recent works, e.g., context-aware query rewriting and RL-optimized QR. Besides, the improvements over BoN are not that impressive (less than 5%).

**Questions:**

1. The authors should show results under an independent evaluator to rule out judge-specific optimization.

2. The authors should add broader baselines beyond BoN (e.g., context-aware QR, RL-optimized QR) to demonstrate improvements over stronger baselines.

---

### Note · Authors · 2025-11-17

I have read and agree with the venue's withdrawal policy on behalf of myself and my co-authors.